# Attempts to Understand Oral Mucositis in Head and Neck Cancer Patients through Omics Studies: A Narrative Review

**DOI:** 10.3390/ijms242316995

**Published:** 2023-11-30

**Authors:** Erin Marie D. San Valentin, Kim-Anh Do, Sai-Ching J. Yeung, Cielito C. Reyes-Gibby

**Affiliations:** 1Department of Emergency Medicine, The University of Texas MD Anderson Cancer Center, Houston, TX 77030, USA; 2Department of Interventional Radiology, The University of Texas MD Anderson Cancer Center, Houston, TX 77030, USA; 3Department of Biostatistics, The University of Texas MD Anderson Cancer Center, Houston, TX 77030, USA

**Keywords:** head and neck cancer, oral mucositis, pain, genomics, transcriptomics, microbiomics

## Abstract

Oral mucositis (OM) is a common and clinically impactful side effect of cytotoxic cancer treatment, particularly in patients with head and neck squamous cell carcinoma (HNSCC) who undergo radiotherapy with or without concomitant chemotherapy. The etiology and pathogenic mechanisms of OM are complex, multifaceted and elicit both direct and indirect damage to the mucosa. In this narrative review, we describe studies that use various omics methodologies (genomics, transcriptomics, microbiomics and metabolomics) in attempts to elucidate the biological pathways associated with the development or severity of OM. Integrating different omics into multi-omics approaches carries the potential to discover links among host factors (genomics), host responses (transcriptomics, metabolomics), and the local environment (microbiomics).

## 1. Introduction

Head and neck squamous cell carcinoma (HNSCC) is the seventh [1] most common type of cancer worldwide. There were an estimated 54,000 new cases in the United States in 2022 [2]. The treatment of HNSCC is based largely on the primary tumor location and stage of the disease. Early-stage disease is treated with single-modality treatment, and advanced-stage disease is treated with multi-modal therapy. The majority of HNSCC patients present with loco-regionally advanced disease. The treatment modalities for HNSCC with curative intent are surgery and/or radiotherapy with or without concurrent systemic therapy. However, treatment-related toxicities are a significant concern. In 2016, the American Cancer Society guidelines [3] for HNSCC underscored the need to recognize the potential late and long-term complications or toxicities of cancer treatment, as well as its undertreatment and management. 

Oral mucositis (OM) is a painful and debilitating treatment-related toxicity in HNSCC patients. OM is a common side effect of cancer treatment, particularly in patients with HNSCC who undergo radiation therapy and/or chemotherapy. The mucous membranes of the mouth and throat are highly sensitive to radiation and chemotherapy, which can damage the cells and cause inflammation. The incidence of OM caused by chemotherapy or radiotherapy can be very high, ranging from 40% to 90% [4,5].

Radiotherapy plans for HNSCC may be administered via fractionated or bolus therapy [6], and each approach has its own distinct effects both at the cellular and molecular levels. Among those receiving radiotherapy, a dose–response relationship is observed, with OM appearing after a cumulative dose of 30 Gy [7,8]. Patients with OM complain of painful mouth ulcers and difficulty in eating or swallowing [9,10], leading to dose reduction or interruption of treatment, increased opioid consumption, visits to the emergency department (ED), and hospitalization [11].

While most patients develop OM, studies show individual variability in the severity and persistence despite the receipt of similar cancer treatments for HNSCC. In a recent study, up to 90% of patients developed OM, but only 36% developed severe OM [12]. Longitudinal patterns of OM also vary, with some patients experiencing early resolution [13,14,15], whereas others develop chronic [16] OM. Epidemiological, behavioral, and clinical variables explain some of the variation observed in OM. Figure 1 shows factors associated with OM, i.e., older age, gender, oral hygiene, total radiation dose, smoking, systemic diseases, radiotherapy technique, combined chemoradiation, ICI, malnutrition or cachexia, and lack of antibiotic use at the early stage. However, evidence suggests these factors only explain some of the variations observed in OM. With advances in molecular technology, studies have explored biomarkers for identifying patients at risk for severe OM and assessing potential biological mechanisms of this complex trait.

## 2. “Omics” Approaches May Help in Risk Assessment and Developing Personalized and More Effective Therapies OM

Because of the increasing accessibility and decreasing costs of high-throughput molecular advancements, it is now possible to assess biomarkers that measure events at the physiological, cellular, and molecular levels and utilize them to aid in identifying individuals at risk, as well as establishing predictive models for severe and persistent treatment-related toxicities.

OM may be a classic example of gene–environment interaction, requiring an initiating event (chemotherapy, radiation, or targeted therapy) and host genetic susceptibility. As a complex human trait, it is expected that multiple genes underlie the development of OM. In order to further understand the mechanism, risks, and management of OM, utilizing and integrating other omics approaches for OM is an ideal strategy. Furthermore, assessing the interaction of these genes with epidemiological, behavioral, and clinical variables will provide an avenue to identify novel markers and tools to improve the risk prediction of OM.

In this narrative review, we have used the database Ovid MEDLINE, using major search concepts including mucositis, (genome, microbiome, proteome, transcriptome) omics and cancer. We used both medical subject headings (MeSH terms) and free text words for relevant search concepts. The search limits we used are studies in human and publication year from 2003 to 2022 (past 20 years). The search returned 425 results. From these 425 results, we did initial screening and selected 235 results (Appendix A).

We subsequently manually searched the reference lists of the selected articles and of related review articles to identify additional relevant studies for inclusion, including animal models. From the selected human studies, we retrieved articles focusing on head and neck cancer patients published between January 2015 and October 2022. Because animal OM models have also been useful in understanding the pre-clinical potential of pharmacological targets for OM, studies of animal OM models were also subsequently included for our review of metabolomics and transcriptomics. The literature search and information extraction were conducted by CCRG and EMSV. We highlight studies conducted from 2015 to 2022 to identify up-to-date putative biomarkers for OM, for predicting the risk, severity of OM, and identifying novel therapeutic targets through different omics approaches.

### 2.1. Genomics

To date, candidate gene, pathway-based, and genome-wide association approaches have been used to identify host genetic susceptibility for OM. Candidate gene studies focus on specific genes of interest that have already been previously selected based on a prior knowledge or hypothesis, aiming to determine the potential role in a specific trait or disease [17]. In contrast, pathway-based studies consider a set of genes that may be involved in common biological pathways or functions [18], thus allowing a collective impact on a trait. Genome-wide association studies, or GWAS, comprehensively scan the whole genome to uncover genetic variants associated with a trait, without any prior assumptions about the specific genes or pathways that may be involved [19]. In Table 1, we have summarized the genes and single nucleotide polymorphisms (SNPs) in relation to OM in HNSCC patients and identified them using different genomic approaches.

#### 2.1.1. Candidate-Gene Studies

Studies have demonstrated associations between gene polymorphism, DNA repair function, and sensitivity to radiotherapy. Using the search terms “SNPs and mucositis” and “gene(s) and mucositis”, Reyes-Gibby et al. [20] conducted a literature search of human studies of OM published before 2016. Their review of the literature identified 27 genes across the various studies, with the most commonly cited genes involved in methylation, DNA synthesis, and DNA repair mechanisms, i.e., X-ray repair cross-complementing gene 1 (*XRCC1*), and excision repair cross complementation group 1 (*ERCC1*) [20].

A subsequent review of the literature for the period 2015–2022 showed studies focusing on HNSCC patients who received radiation therapy. Five studies were conducted on a small number of HNSCC patients receiving radiation therapy in the oncology department at the Medical University of Lublin (Lublin, Poland). They showed that polymorphisms in *GHRL* (rs1629816) [21], *TNFRS1A* [22], (rs4149570, rs767455) [23]), *TNFA* (rs1799964) [24], and *APEH* (rs4855883) [25] were significantly associated with OM. In a larger sample (*n* = 114) of Chinese patients with HNSCC receiving radiotherapy, Chen et al. [26] found the *XRCC1* variant (rs25487) to be associated with an increased risk of OM.

#### 2.1.2. Pathway-Based

Although comprehensive, the pathway-based approach relies on a priori knowledge of SNPs, gene functions, and biological plausibility. Reyes-Gibby et al. [20] used a pathway-based approach for OM. They first reviewed literature before 2015 on genetic association studies of OM. They found 27 genes from 28 published studies. Using the 27 genes, they generated gene networks for OM using Ingenuity Pathway Analysis (IPA), which is a bioinformatics tool using the Ingenuity Knowledge Base, a comprehensive database containinginformation on biomolecules and their relationships. They found *TP53*, *CTNNB1*, *MYC*, *RB1*, *P38 MAPK*, and *EP300* as the most biologically significant molecules and the uracil degradation II (reductive) and thymine degradation pathways as the most significant biological pathways. They then conducted a genetic association study for OM in 885 HNSCC patients (with OM = 186; without OM = 699) utilizing 66 SNPs within the eight most connected IPA-derived candidate molecules. The top-ranked gene identified through this association analysis was RNA-binding proteins (RBP) (rs2227311, *p*-value = 0.034, odds ratio = 0.67). To date, there has been a limited application of the pathway-based approach in the study of OM.

**Table 1 ijms-24-16995-t001:** Summary of identified genes and corresponding SNPs that were found to be implicated in oral mucositis in head and neck cancer patients.

Year	First Author	Approach	Sample Size	Therapy	Sample	Phenotype	Genes	SNPs
2022	Schack [27]	Genome-wide	Discovery = 1183 Danish Cohort Replication = 597 Danish Cohort Validation = 235 Asian Cohort	RTx	Buffy coats	Mucositis 0: no 1: erythema 2: patchy 3: confluent 4:ulceration	*STING1*	rs1131769
2020	Mlak [23]	Candidate gene	60	RTx	Peripheral blood	Mucositis RTOG/EORTC	*TNFRS1 A*	rs767455
2020	Mlak [24]	Candidate gene	62	RTx	Peripheral blood	Mucositis RTOG/EORTC	*TNF alpha*	rs1799964
2020	Yang [28]	Genome-wide	960 560	RTx	Blood	RTOG/EORTC	*TNKS*	rs117157809
2018	Brzozowska [25]	Candidate gene	62	RTx	Peripheral blood	Mucositis RTOG/EORTC	*APEH*	rs4855883
2018	Brzozowska [22]	Candidate gene	58	RTx	Peripheral blood	Mucositis RTOG/EORTC	*TNFRS1 A*	rs4149570
2018	Brzozowska [21]	Candidate gene	65	RTx	Peripheral blood	Mucositis RTOG/EORTC	*GHRL*	rs1629816
2017	Chen [26]	Candidate gene	114	RTx	Peripheral blood	Mucositis RTOG/VRS	*XRCC1*	rs25487
2017	Reyes-Gibby [20]	Pathway-based	885	RTx and/or CTx	Peripheral blood	Oral Mucositis (ICD)	*RB1*	rs2227311

RTx: Radiotherapy; CTx: Chemotherapy; RTOG: Radiation Therapy Oncology Group; WHO: World Health Organization; EORTC: European Organization for Research and Treatment of Cancer; ICD: International Classification of Diseases.

#### 2.1.3. Genome-Wide Approach

Hypothesis-free, whole genome-wide association studies are effective for identifying genetic factors contributing to complex diseases. However, there has been little use of this approach in research because of the cost and the need for very large sample sizes for replicable results. A study by Schack et al. [27] utilized three cohorts (discovery phase: 1183 Danes, replication phase: 597 Danes, and validation phase: 235 Asians) and found the rs1131769 of the STimulator of Interferon Response cGAMP Interactor 1 gene (*STING1*) to be significantly associated with OM. STING1 (also known as transmembrane protein 173) has been associated with infection, inflammation, immunity, autophagy, and cell death [29,30,31,32]. An earlier genome-wide study by Yang et al. [28] found SNP rs117157809 in the protein-coding gene TNKS (Tankyrase) associated with more than three-fold OM risk in patients with nasopharyngeal cancer. TNKS plays a role in radiation-induced damage. Depletion of TNKS is associated with increased sensitivity to ionizing radiation-induced mutagenesis, chromosome aberration, telomere fusion, and cell killing [33].

### 2.2. Transcriptomics and Proteomics

Downstream of genome expression, a collection of RNA molecules, called the transcriptome, is derived from the protein-coding genes. These RNA molecules direct the synthesis of the final product of genome expression, the proteome, the cell’s repertoire of proteins, which specifies the nature of the biochemical reactions that cells are able to carry out. Transcriptome analysis typically investigates differential gene expressions that occur in different (e.g., normal vs. abnormal) states. Transcriptomic tools make use of high-throughput technologies such as microarrays, RNA sequencing, and, more recently, single-cell transcriptomics, wherein an individual cell is profiled. Transcriptomic data aid in the elucidation of the mechanisms involved in OM. By analyzing patterns of gene expression, dysregulated key pathways and biomarkers can be identified. For example, microRNAs (miRNA) are small non-coding RNAs that play an important role in post-transcriptional gene regulation by binding to specific mRNA molecules and initiating degradation or translational inhibition, thereby affecting gene expression patterns. In the context of OM, the miR-1206 variant has been associated with a three-fold increase in the risk of developing methotrexate-induced OM [34], whereas miR-200c showed promising results in reducing ROS production and repressing proinflammatory cytokines in animal models [35].

While transcriptomics and proteomics provide different types of data, there is a significant overlap between the two approaches, since changes in gene expression may lead to changes in protein expression, and changes in protein expression can also regulate other downstream gene expression and protein levels. By integrating data from both approaches, we can gain a more complete understanding of the biological processes, networks, and pathways involved in OM and identify targets for the treatment or prevention of therapy-induced OM.

The proteomic approach for OM research has focused on biomarker development, particularly on inflammatory proteins observed after treatment-induced OM. Kiyomi et al. [36] utilized a bead array and an enzyme-linked immunosorbent assay (ELISA) to identify potential biomarkers of OM from saliva or oral swab samples were taken from 20 leukemia or head and neck cancer patients undergoing treatment. Their study suggests that salivary IL-6, IL-10, and TNF-α may serve as predictors of OM occurrence and grade. Additionally, Jehmlich et al.[37] were able to identify 48 (see Table 2) unique proteins that differ significantly between OM and non-OM groups with saliva and/or oral swabs obtained from a pool of 50 head and neck cancer patients. Among those identified is proteinase 3 (PRTN 3), a secretory multifunctional serine protease that can degrade elastin, fibronectin, and collagen. PRTN3 is released upon neutrophil activation and degranulation during tissue injury inflammation. This has also been implicated in the serum proteomic profile of HNSCC patients. While different protein profiles were obtained from the patients due to variations in tumor status and collection time points, most of the 48 proteins are extracellular, have important roles in inflammatory and innate immune responses, and complement activation cascade [37].

### 2.3. Metabolomics

Metabolomics involves the comprehensive characterization of low-molecular-weight molecules, metabolites, and metabolism in biological systems. Unlike genomics and proteomic strategies, metabolomics aims to measure molecules that have disparate physical and chemical properties [38], which make it more challenging to study. To address this, the metabolome is investigated using different analytical methods and platforms. Studies have collected saliva, serum, and volatile organic compounds to reflect various pathological conditions, making it an attractive source for the diagnosis of systemic diseases and potential biomarkers of pathological conditions. Analytical platforms such as liquid chromatography-mass spectrometry (LC-MS), gas chromatography-mass spectrometry (GC-MS), capillary electrophoresis time-of-flight mass spectrometry (CE-TOF-MS), and nuclear magnetic resonance (NMR) spectroscopy are commonly used to measure and profile small molecule metabolites [39]. In OM research, metabolomics provides valuable insights into the biochemical changes that occur in the oral mucosa due toradiation or chemotherapy treatment.

Yatsuoka et al. (2021)[40] analyzed the time course of salivary metabolic profiles using CE-TOF-MS in nine male patients with HNSCC who received radiation therapy of at least 50 Gy. Partial least squares regression-discriminant analyses showed that histidine and tyrosine highly discriminated between high-grade OM and low-grade OM at baseline. Moreover, γ-aminobutyric acid (GABA) and 2-aminobutyric acid (2AB) concentrations were higher in the high-grade OM group than in the low-grade OM group. While GABA is known to be correlated with stress levels, 2AB is known to increase under high oxidative stress conditions, which can be induced by radiation.

The metabolomic changes brought about by maxillofacial and oral massage (MOM) attempting to attenuate severe radiotherapy-induced OM in patients with nasopharyngeal carcinoma were also explored by Yang et al. [41]. They identified enhanced levels of the metabolites 9S-HEPE and 15-HETE among patients receiving MOM after radiotherapy, both of which are known to play a role in inhibiting inflammatory responses through different pathways [42].

Animal OM models have also been useful in understanding the pre-clinical potential of pharmacological targets. Two independent studies involving the use of Chinese herbal medicine, Shuanghua baihe tablet (SBT) and Kouyanqing granules (KQG), have used OM rat models to determine the effects in alleviating OM symptoms and elucidate the potential metabolic pathways involved. Geng et al. (2021) [43] showed the role of SBT in metabolic-related pathways such as linoleic acid and cholic acid metabolism to alleviate the inflammatory symptoms of OM. KQG, on the other hand, shows promising effects by attenuating symptoms of oral ulcers through regulation of the neuroimmunoendocrine system, oxidative stress, and tryptophan metabolism [44].

Metabolomics data have the potential to highlight metabolic pathways that contribute to OM pathogenesis, including specific amino acids and lipid metabolites that could reveal potentially novel therapeutic targets for OM. Moreover, the improvements in analytical methods for metabolomic analysis allow for the identification of thousands of metabolites that could be correlated to the clinical phenotypes of OM to improve risk prediction and assessment.

**Table 2 ijms-24-16995-t002:** Omics approaches and putative molecular markers: animal models and head and neck cancer patients.

Year	First Author	Phenotype	Samples	Therapy	Sample Size	Methods	Targets	Results
	Metabolomics							
	Animal Models							
2021	Geng [43]	Mucositis (0–5)	Serum from OM rat model, induced with 5-FU and 10% acetic acid	CTx	30 rats	UHPLC	Cholic acid, linoleic acid, 4-pyridoxic acid, LysoPC	Shuanghua Baihe tablets improve inflammatory symptoms of oral mucositis.
2020	Chen [44]	Induced oral ulcers and degree of healing	Serum from OM rat model, induced with 15% chloral hydrate	CTx	42 rats	LC-QTOF/MS	5-HT, GABA	Kouyanqing granules attenuate the symptoms of oral ulcers worsened by sleep deprivation through regulation of the neuroimmunoendocrine system, oxidative stress levels, and tryptophan metabolism.
	Clinical Samples							
2021	Yatsuoka [40]	NCI CTCAE	Saliva	CTx/RTx	9 HNC	CE-TOF-MS	tryptophan, D-glucose, D-glutamate, GABA, 2-AB	Pre-treatment concentrations of gamma-aminobutyric acid and 2-aminobutyric acids were higher in the high-grade OM group.
2021	Yang [41]	NRS 010	Peripheral blood	RCTx	10 NPC	UHPLC-MS/MS	9-HEPE, 15-HETE	MOM promotes the release of anti-inflammatory lipids to reduce tissue damage; enhancement of 9S-HEPE and 15-HETE in all radiation doses.
	Microbiomics							
2020	Reyes-Gibby [12]	NCI CTCAE	buccal mucosal	RTx and/or CTx	66 Locoregional HNSCC	16S rRNA	*Cardiobacterium*, *Granulicatella*, *Prevotella*, *Fusobacterium*, *Streptococcus*, *Megasphaera*, *Cardiobacterium*	Genera abundance was associated with the hazard for the onset of severe OM.
2020	Vesty [45]	WHO	saliva and oral swabs	RTx	19 HNC	NGS	*Fusobacterium*, *Haemophilus*, *Tannerella*, *Porphyromonas and Eikenella*, *Candida*	Gram-negative bacteria on the buccal mucosa may influence susceptibility to developing OM.
2019	Subramaniam and Muthukrishnan [46]	WHO	unstimulated whole saliva	RTx and RCTx	24 HNSCC	16S rRNA	*Staphylococcus aureus*, *Staphylococcus epidermidis*, *Pseudomonas aeruginosa*, *Escherichia coli*, *Klebsiella pneumoniae*	The bacterial isolates obtained during and at the end of therapy appeared to express a higher level of antibiotic-resistance genes (*VIM2*, *MCR-1*, *TET[K]*, *blaKPC*) than those isolated at the onset of therapy.
2018	Hou [47]	RTOG	oral swabs	RTx	19 NPC	16S rRNA	*Prevotella*, *Fusobacterium*, *Treponema*, *Porphyromonas*	*Prevotella*, *Fusobacterium*, *Treponema* and *Porphyromonas* showed dynamic synchronous variations in abundance throughout the course of radiation therapy, frequently coinciding with the onset of severe mucositis.
2017	Zhu [48]	RTOG	oral or retropharyngeal mucosa swabs	RTx	41 NPC	*1* *6S rRNA*	*Firmicutes*, *Proteobacteria*, *Bacteroidetes*, *Fusobacteria*, *Actinobacteria*, *Spirochaetes*, *Cyanobacteria*, *Verrucomicrobia*, *Acidobacteria*, *TM7*, *Deinococcus-Thermus* and *SR1*	Oral microbiota changes correlate with the progression and aggravation of radiotherapy-induced mucositis in patients with nasopharyngeal carcinoma.
	Microbiota							
2018	Almstahl [49]	WHO	Swab culture	RTx	33 HNC	Culture	*Neisseria*, *Fusobacterium*, *Prevotella*, *Candida*	Levels of *Neisseria* decreased and mucosal pathogens increased during RT; 2 years post-treatment, *Fusobacterium* and Prevotella *decreased;* growth of *Candida* increased
2018	Gaetti-Jardim [50]	NCI CTCAE	Supra and subgingival biofilms	RTx	28 HNC	Culture	*Candida*, *Enterobacteriaceae*	*Candida* and family *Enterobacteriaceae* showed increased prevalence with RT, and were associated with the occurrence of mucositis and xerostomia
	Transcriptomics							
	Animal Models							
2021	Geng [43]	Mucositis (0-5)	Serum from OM rat model, induced with 5-FU and 10% acetic acid	CTx	30 rats	Whole genome sequencing	ALOX15, CYP2J2, CYP1A1, ALOX15, GATM, ALAS2, PLA2G5	Shuanghua Baihe tablets improve inflammatory symptoms of oral mucositis.
2021	Saul-McBeth [51]	Induced oral ulcers, % damage	OM mice model; Induced with head and neck irradiation	RTx	3 mice	RNA Seq	IL-17RA	IL-17RA provides protection during HNI-induced OM by preventing excess inflammation during ulceration phase of OM.
	Clinical Samples							
2018	Mlak [52]	RTOG/EORTC	Plasma	RTx	60 HNC	Microarray	RRM1	RRM1 gene expression in cfRNA allows for estimating risk of severe OM.
	Proteomics							
2015	Jehmlich [53]	NCI CTC v3	Unstimulated whole saliva	RTx	50 HNC	MS	RPL18A, C6orf115, PRTN3, RPS20, FGB, ARPC1B, PLBD1, GGH, ANXA6, FGG, ANP32E, CTSG, PTGR1, SERPINA1, MDH2, CORO1A, HSPE1, BAHCC1 CP, MMP9, GCA, PLYRP1, SCGB2A1, GPI, PPIC, QRDL, HIST1H4A, HNRNPA2B1, ATP5B, LTA4H, TIMP1, TKT, RPL10A, AZU1, MMP8, RPLP2, ARPC4, CAT, S100A8, B2M, SERPING1, CYBB, ELANE, C3, CALML5, ITIHRPS15A, ACTR2	48 proteins differed significantly between OM group and non-OM group. 17 proteins displayed increased levels and 31 proteins decreased in level in OM.

RTx: Radiotherapy; CTx: Chemotherapy; RCTx: Radiochemotherapy; NCI CTCAE: National Cancer Institute Common Terminology Criteria for Adverse Events, NRS: numerical rating scale; WHO: World Health Organization; RTOG: Radiation Therapy Oncology Group; EORTC: European Organization for Research and Treatment of Cancer; 5-FU: 5-fluoruracil; HNC: head and neck cancer; NPC: nasopharyngeal cancer; HNSCC: head and neck squamous cell carcinoma; UHPLC: ultra-high-performance liquid chromatography; LC-QTOF/MS: liquid chromatography- quadrupole time-of-flight (TOF)/mass spectrometer (MS); CE-TOF-MS: capillary electrophoresis-TOF-MS; NGS: next generation sequencing; 5-hydroxytryptamine; GABA: γ-aminobutyric acid; 2-AB: 2-aminobutyric acid; 9-HEPE: 9-hydroxyeicosapentanoic acid; 15-HETE: 15-hydroxyeicosatetraenoic acid; OM: Oral mucositis; MOM: maxillofacial and oral massage.

### 2.4. Microbiomics of OM

The application of microbiomics has been used in OM research in HNSCC patients because the microbiome (the collective community of microorganisms) in the oral cavity has been implicated in the development of OM, given the diversity of microbial colonization in the oral cavity. It is hypothesized that changes in the composition of the oral microbiome can increase inflammation and tissue damage in the oral mucosa. However, anti-bacterial treatment strategies for OM, such as iseganan, have not been successful [54].

These results suggest two scenarios: microbiome changes precede OM or OM influences the microbiome. While a classic “chicken or egg question” arises [55,56], the reality is possibly a combination of both. The relationship between OM and the oral microbiome is likely bidirectional. As the mucosal barrier is compromised in OM, changes in the microbiome can occur due to increased exposure to pathogens and alterations in the local environment. These microbial changes might also contribute to the severity and persistence of mucositis. Although not specific to HNSCC, Hong et al.[57] previously showed that antineoplastic agents such as 5-fluorouracil represent the primary initiators of OM and triggers inflammation. These agents may induce microbiome disruptions, and in turn the dysbiotic microbiome could play a role in the clinical course of lesions aggravating epithelial injury.

Reyes-Gibby et al. [12] identified different features associated with the risk of OM at baseline (*Cardiobacterium*, *Granulicatella)*, immediately before the development of OM *(Prevotella*, *Fusobacterium*, *Streptococcus)*, and immediately before the development of severe OM (*Megasphera*, *Cardiobacterium*). Interestingly, *Prevotella* and *Fusobacterium* have also been identified in the study of Hou et al. [47], where these features showed dynamic synchronous variations in abundance throughout the course of radiation therapy and frequently coincided with the onset of severe OM.

The combination of microbiomic and genomic data may also be a powerful approach to identifying key features for therapy. For example, in 24 HNSCC patients who received radiotherapy and concomitant chemoradiotherapy, microbial species (*Staphylococcus aureus*, *S. epidermis*, *Pseudomonas aeruginosa*, *E. coli*, and *Klebsiella pneumoniae*) from saliva samples expressed higher levels of antibiotic-resistance genes (VIM2, MCR-1, TET(K), blaKPC) after receiving cancer therapy [46]. On the other hand, Xia et al. [58] investigated the protective effects and mechanism of a probiotic cocktail treatment on nasopharyngeal cancer OM rat models induced by chemoradiotherapy. While this study focused on the gut microbiome, they were able to support previous hypothesis that the modulation of the gut microbiota through the probiotic cocktails can, in turn, modulate the response to cancer treatment. Specifically, the probiotics aid in prevention of system-immune activation and inflammation which eventually ameliorates OM.

There are also several challenges in the study of microbiome. For example, the interpretation of microbiome data can be challenging with the utilization of complex tools, a lack of standardization, and highly variable data that are easily influenced by factors such as diet, medications, and even oral hygiene. Moreover, observed changes in the oral microbiome associated with OM are not adequate to establish their causal or modifier roles in OM. Treatment-induced dental caries, hyposalivation, and xerostomia may change the environment in the upper orodigestive tract to affect the composition of the microbiome.

More pertinent to the OM studies is the incorporation of microbiome data that generates additional statistical challenges. The observed Microbiome Data Structure from either 16S or whole metagenomic sequencing profiling can be summarized as a matrix X of counts for each taxonomic feature (OTU, ASV, or SGB) or functional activity in each sample.

Some aspects of microbiome data structure are shared with that of other high-throughput data types: in particular, bulk RNA-sequencing data are also compositional and count-based, while single-cell RNA sequencing data share many similarities with microbiome data, including the presence of many exact zero values. Methods developed for the analysis of these data types may sometimes be applicable to microbiome data as well, although in many cases microbiome-specific methods may be preferred. In particular, since many microbiome features are rare or zero-inflated, taking the tree relations among features into account can be beneficial in statistical inference. Several studies in recent times have linked gut microbiome (GM) diversity to the pathogenesis of cancer and its role in disease progression through immune response, inflammation, and metabolism modulation. Jiang et al. [59] overviewed the most important network methods for integrative analysis, emphasizing on methods that have been applied or have great potential to be applied to the analysis of multi-omics integration of microbiome data. They compared advantages and disadvantages of various statistical tools, assessed their applicability to microbiome data, discussed their biological interpretability, and highlighted on-going statistical challenges and opportunities for integrative network analysis of microbiome data. Several tools and approaches have been employed to analyze multi-omic interactions in different diseases, including applied network analysis [60], weighted gene co-expression network analysis (graph-regularized vector autoregressive model [61], DiffCoEx20 [62], principal component analysis and DESeq2 in combination with bioinformatics databases Gene Ontology (GO), Kyoto Encyclopedia of Genes and Genomes (KEGG), and Reactome. In the context of OM, Bruno et al. [56] collected the latest articles indexed in the PubMed electronic database, analyzed the bacterial shift through 16S rRNA gene sequencing methodology in cancer patients under treatment with oral mucositis, assessed whether changes in the oral and gut microbiome causally contributed to OM, and explored the emerging role of a patient’s microbial fingerprint in OM development and prediction.

A very important resource is the National Microbiome Data Collaborative (NMDC), a new US-based pilot initiative launched in June 2019 to support microbiome data exploration and discovery through a collaborative, integrative data science ecosystem. The primary goal is to democratize microbiome data science by providing access to multi-omics microbiome data to support reproducible, cross-study analyses aligned with the FAIR (Findable, Accessible, Interoperable and Reusable) data principles. Finally, the most recent comprehensive review by Peterson et al. [63] conducted a study of statistical methods for the analysis of microbiome data, discussing visualization & exploratory analysis, differential abundance analysis, regression modeling, microbiome network inference methods, and integration of microbiome with other omics data.

### 2.5. Challenges in Integrating a Multi-Omics Approach for OM

OM is a dynamic and intricate condition influenced by genetic factors, the environment, and treatment regimen employed. Existing literature indicates that OM development may result from genetic mutations in both germline and somatic sources [64], with their interactions implicated in the development, severity, and progression of the disease. This makes data integration and analysis challenging due to the complexity and heterogeneity of phenotypes and data arising from gene–gene interactions, somatic and germline gene expression relationships, and the interaction of the different risk factors that determine OM risk and prognosis. As summarized in Table 2, individual omic fields already provide novel prognostic information of OM risk; however, no single data set can explain the entirety of risk prediction as these fields provide complex information [65]. Integration of these data sets with the previously identified traditional risk factors provide potential for multi-omics studies to uncover powerful tools and potential precision medicine approaches for OM [64]. However, the data sets generated from these studies are often complex, heterogenous, and high-dimensional. There is a continuous need to standardize methodologies, and more importantly, develop and refine computational and statistical techniques and resources to harmonize datasets from varied assay types and data modalities.

Statistical methods are ubiquitous in the integrative analysis of the genomics, microbiome, metabolomics, proteomics, and transcriptomics data to effectively conduct risk evaluation and developing more effective treatment in OM. In the past, a single-level analysis has been extensively conducted, where the omics measurements at different levels, including mRNA, microRNA, CNV, and DNA methylation, were analyzed separately. Integrative analysis offers an effective way to borrow strength across multi-level omics data and can be more powerful than single-level analysis. Integration of diverse genomic data from many platforms has the potential to increase precision, accuracy, and statistical power for the identification of combinations of important biomarkers associated with clinical outcomes. During the early efforts to examine the utility of genomic data integration, many investigators, including Kim-Anh Do and her team, have studied this challenge as published previously [66,67,68,69] using full Bayesian methods to investigate pairwise interactions between mRNA and miRNA expression, and mRNA expression and DNA methylation. Daemen et al. [70] and Wu et al. [71] developed a two-step frequentist method that combines mRNA expression and proteomic data, but few have truly combined information from three or more molecular platforms, as investigated by Chekouo et al. [72]. A recent comprehensive review conducted by Wu et al. [73] focused on variable selection methods for integrative analyses, as well as existing supervised, semi-supervised and unsupervised integrative analyses within parallel and hierarchical integration studies, respectively.

## 3. Conclusions

The different types of “omics” studies provide preliminary evidence that high-throughput methodologies applied to study different aspects of the host (host biomarkers), response to cancer treatment (chemotherapy and/or radiotherapy), and microbiological factors (microbiome and infection) are feasible. There is a paucity of these studies, and there is a complete lack of studies that integrate and analyze multi-omics data to examine the importance of genomic variations, gene–environment interactions, and mechanisms of the host response to a chemical or radiational cytotoxicity to the development of OM. Moreover, the addition of targeted therapy to the mix of antineoplastic treatment for HNSCC adds further complexity into the pathogenic mechanism of OM in HNSCC patients.

## 4. Future Directions

The basic principle of molecular epidemiology [74] is that neither genetics nor environment alone are responsible for individual variation in disease presentation and severity. Whereas traditional field-based epidemiological approaches have identified subgroups at higher risk for OM (older age, body mass index, radiotherapy dose), the development of high throughput molecular laboratory techniques has allowed for the use of biological markers in disease prevention and risk prediction. These biomarkers measure events at the physiological, cellular, and molecular levels, thus improving our understanding of the epidemiology of diseases. Integrating the use of molecular methods of analyses allow for a better understanding of biological mechanisms and improves assessments of individual risks by providing person-specific information (a genetic profile, etc.) along with clinical information. Such tools could identify subgroups who might benefit most from intervention and contribute to developing personalized and more effective therapies while reducing toxic side effects.

Applying omics approaches may potentially identify subgroups of patients who will benefit from a specific intervention or treatment. However, a common limitation of the studies was the heterogeneity of the population sample, the small sample size, the use of different OM measures, and the retrospective study design.

Challenges include the heterogeneity of the underlying pathogenic mechanisms of OM given that there are now three main types of cancer treatment for HNSCC (chemotherapy, radiotherapy, and targeted therapy) that can cause OM. Prospective studies that will allow assessment of the timing of OM onset and the response to conventional mucositis treatments are needed to identify pathogenic mechanisms. In addition, the clinical features of OM can help identify the involvement of infection (viral, fungal, or bacterial) or immune reaction, i.e., lesion appearance, location, redness, swelling, ulceration, pain severity, etc. The use of statistical approaches to assess which clinical features of OM will correlate with different multi-omics patterns will provide insight into the pathogenic mechanisms and potential treatment targets.

Importantly, omic studies are resource-intensive and therefore, it is important to note that funding announcements by the US National Institute of Dental and Craniofacial Research (NIDCR) and the US National Cancer Institute (NCI) may provide the opportunities to pursue multi-center studies applying omics approaches to OM. In particular, a reissue by NIDCR of PAR-17-154 [73] calls for prospective observational designs and biomarker validation studies. Considered appropriate would be epidemiologic studies of disease prevalence or incidence, cohort studies prospectively ascertaining risk factors for disease development, cohort studies that provide longitudinal follow-up of treatment outcomes, case–control studies with longitudinal follow-up, and large cross-sectional studies or case–control studies evaluating genomic changes, gene-environment interactions, or disease/treatment mechanisms through omics, cellular, and imaging analyses. For the biomarker validation studies, the reissue of PAR-17-154 will [75] promote advanced analytic and/or clinical validation of strong candidate biomarkers and endpoints for the diagnostic or prognostic utility to demonstrate that biomarker or endpoint change is reliably correlated with pathophysiology, clinical outcome, therapeutic target engagement or treatment response.

## Figures and Tables

**Figure 1 ijms-24-16995-f001:**
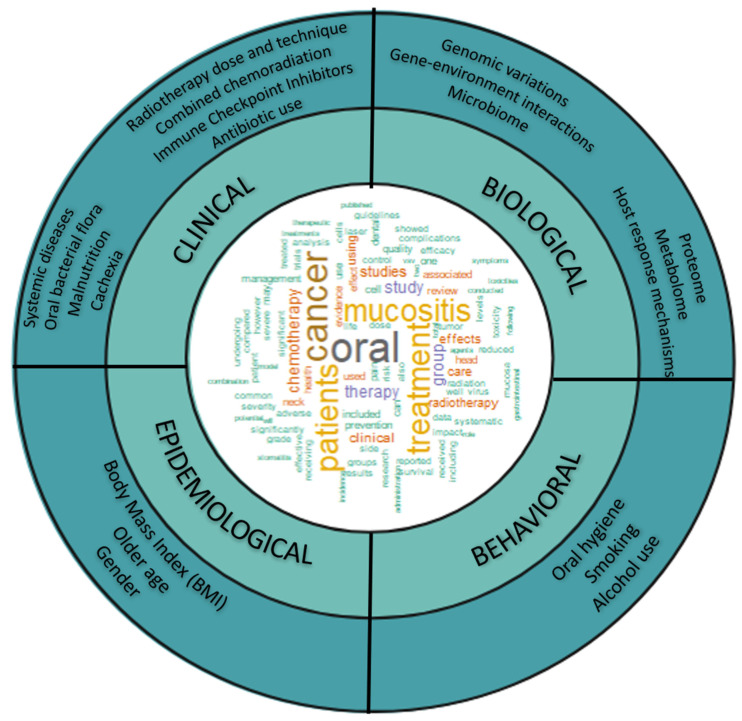
Oral mucositis phenotype is heterogenous and known to vary by epidemiological clinical, and behavioral factors and molecular markers of risk. Word cloud showing frequency of appearance of terms related to oral mucositis based on the abstract of publications for oral mucositis.

## Data Availability

Data sharing not applicable.

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
