# Peer review of "Attempts to Understand Oral Mucositis in Head and Neck Cancer Patients through Omics Studies: A Narrative Review"

_ijms, 2023, doi:10.3390/ijms242316995_

Round 1

Reviewer 1 Report

Comments and Suggestions for Authors

Review of “Attempts to Understand Oral Mucositis in Head and Neck Cancer Patients Through Omics Studies: A Narrative Review”

Oral mucositis in Head and Neck cancer (HNC) patients is a frequent issue, mainly during radio- and chemo- therapy.

In particular, oral mucositis worsens the patient’s quality of life and increases the risk of suspension of therapy, preventing the completion of treatment, and thus potentially reducing survival and disease control. The authors present an exhaustive narrative review of the use of Omics in the understanding of oral mucositis. In particular, they discuss how Omics has been used so far to predict the risk of oral mucositis in patients with HNC.

I believe this could be of help to better select patients who could avoid specific treatments (as adjuvant radiotherapy when not strictly needed) or in which particular precautions should be taken to avoid a premature interruption of treatments.

The review is very well written and structured. Each Omic is separately discussed, and each paragraph can be a starting point for future research in the field.

Author Response

Thank you very much for your thoughtful and encouraging comments on our review. We are grateful for your recognition of the potential for each Omic discussed in our review to serve as a foundation for future investigations. We appreciate your support and look forward to contributing further insights to advance the understanding and management of oral mucositis in HNC patients. If you have any additional suggestions or specific areas you believe warrant further exploration, we would be keen to hear your thoughts. Thank you once again for your valuable feedback.

Reviewer 2 Report

Comments and Suggestions for Authors

Authors have summerized the background of oral mucositis in HNSCC based on the genomics, transcriptomics, proteomics and metabolomics. Here are some comments for authors:

1) It is hard to understand the selection of time periods 2003-2022 and 2015-2022, as in the text there are also reviews before 2015 that are commented. The same difficulty has to be overcame with the selection of human and animal models, as it is not expressed at the supplementary flow diagram.

2) Papers related to other chemotherapy drugs-induced OM and omics should me commented in the text to compare the differences in omics-pathways. Examples: Hong et al. Chemotherapy-induced oral mucositis is associated with detrimental bacterial dysbiosis. Microbiome. 2019 Apr 25;7(1):66. doi: 10.1186/s40168-019-0679-5. PMID: 31018870; PMCID: PMC6482518.

Xia et al. Phase II Randomized Clinical Trial and Mechanistic Studies Using Improved Probiotics to Prevent Oral Mucositis Induced by Concurrent Radiotherapy and Chemotherapy in Nasopharyngeal Carcinoma. Front Immunol. 2021 Mar 24;12:618150. doi: 10.3389/fimmu.2021.618150. PMID: 33841399; PMCID: PMC8024544.

3) The results of the review are between paragraphs with explanation of technics and commenting their quality. Do the authors believe that all these descriptions are necessary?  

Author Response

1) It is hard to understand the selection of time periods 2003-2022 and 2015-2022, as in the text there are also reviews before 2015 that are commented. The same difficulty has to be overcame with the selection of human and animal models, as it is not expressed at the supplementary flow diagram.

Response: We initially screened studies from the past 20 years, thus the 2003-2022 timeframe. We then retrieved recent studies (2015-2022) relating to head and neck squamous cell carcinoma to highlight putative markers for OM. While we also references studies older than 2015, our main analyses utilized studies from 2015-2022. We have edited the paragraph on literature search to improve readability (page 3, lines 81-91)

2) Papers related to other chemotherapy drugs-induced OM and omics should me commented in the text to compare the differences in omics-pathways. Examples: Hong et al. Chemotherapy-induced oral mucositis is associated with detrimental bacterial dysbiosis. Microbiome. 2019 Apr 25;7(1):66. doi: 10.1186/s40168-019-0679-5. PMID: 31018870; PMCID: PMC6482518.

Xia et al. Phase II Randomized Clinical Trial and Mechanistic Studies Using Improved Probiotics to Prevent Oral Mucositis Induced by Concurrent Radiotherapy and Chemotherapy in Nasopharyngeal Carcinoma. Front Immunol. 2021 Mar 24;12:618150. doi: 10.3389/fimmu.2021.618150. PMID: 33841399; PMCID: PMC8024544.

Response: Thank you for suggesting these literature. We have focused our analyses on OM in HNSCC patients and the oral microbiota. However, as findings from the abovementioned literature are also valuable in comparing the differences in omics-pathways, we have referenced them in Microbiomics section, page 10, lines 8-18 and 31-37.

3) The results of the review are between paragraphs with explanation of technics and commenting their quality. Do the authors believe that all these descriptions are necessary?  

Response: We believe the need to elaborate on the different techniques because currently, there is no standardized procedure for each of the omics methodologies. While we generate multitude of data to understand OM in a multi-omics approach, it is a challenge to integrate and analyze from many platforms. Elaborating on these techniques help us understand the best methods and combinations thereof to efficiently predict or treat therapy-induced OM.